# VideoMV: Consistent Multi-View Generation Based on Large Video Generative Model

## Abstract

Generating multi-view images based on text or single-image prompts is a central topic in 3D content creation. Two fundamental questions on this topic are what data we use for training and how to ensure multi-view consistency. This paper introduces a novel framework that makes fundamental contributions to both questions. Unlike leveraging images from 2D diffusion models for training, we propose a dense consistent multi-view generation model that is fine-tuned from off-the-shelf video generative models. Images from video generative models are more suitable for multi-view generation because the underlying network architecture employs a temporal module to enforce frame consistency. Moreover, the video data sets used to train these models are abundant and diverse, leading to a reduced train-finetuning domain gap. To enhance multi-view consistency during generation, we introduce a *3D-Aware Denoising Sampling* procedure, which first employs a feed-forward reconstruction module to get an explicit global 3D model, and then adopts a sampling strategy that effectively involves images rendered from the global 3D model into the denoising sampling loop to improve the multi-view consistency of the final images. As a by-product, this module also provides a fast way to create 3D assets represented by 3D Gaussians within a few seconds. Our approach can generate 24 dense views and converges much faster in training than state-of-the-art approaches (4 GPU hours versus many thousand GPU hours) with comparable visual quality and consistency. By further fine-tuning, our approach outperforms existing state-of-the-art methods in both quantitative metrics and visual effects.

## 1 Introduction

The creation of 3D content plays a crucial role in virtual reality, the game and movie industry, 3D design, etc. However, the scarcity of large-scale 3D data and the high time consumption of acquiring them pose significant obstacles in learning a strong 3D prior from them directly for high-quality 3D content creation. To address the data issue, recent advances, such as DreamFusion Poole et al. (2022) leverage **2D generation priors** learned from large-scale image data to optimize different views of the target object. Despite generating realistic views, such approaches suffer from the multi-face janus problem caused by the lack of the underlying 3D model when learning from images generated by 2D diffusion models. Recent approaches, including MVDream Shi et al. (2023b) and Wonder3D Long et al. (2023), use the attention layers learned from limited 3D data Deitke et al. (2022) to boost multi-view consistency in the generated images. However, these approaches still present noticeable artifacts in multi-view inconsistency and show limited generalizability.

We argue that there are two key factors to achieve high-quality and multi-view consistent image generation results. The first is what data and model we use for pre-training. They dictate the type of feature being learned, which is important for multi-view consistency. The second factor is how to infer an underlying 3D model, which is the most effective way to enforce multi-view consistency.

This paper introduces VideoMV, a novel approach that makes important contributions to both factors. The key idea of VideoMV is to learn **video generation priors** from object-central videos. This approach has three key advantages. First, the data scale of object-central videos is large enough to learn strong video generation priors. Second, video generative models have strong attention modules across the frames, which are important for multi-view consistency Shi et al. (2023b); Long

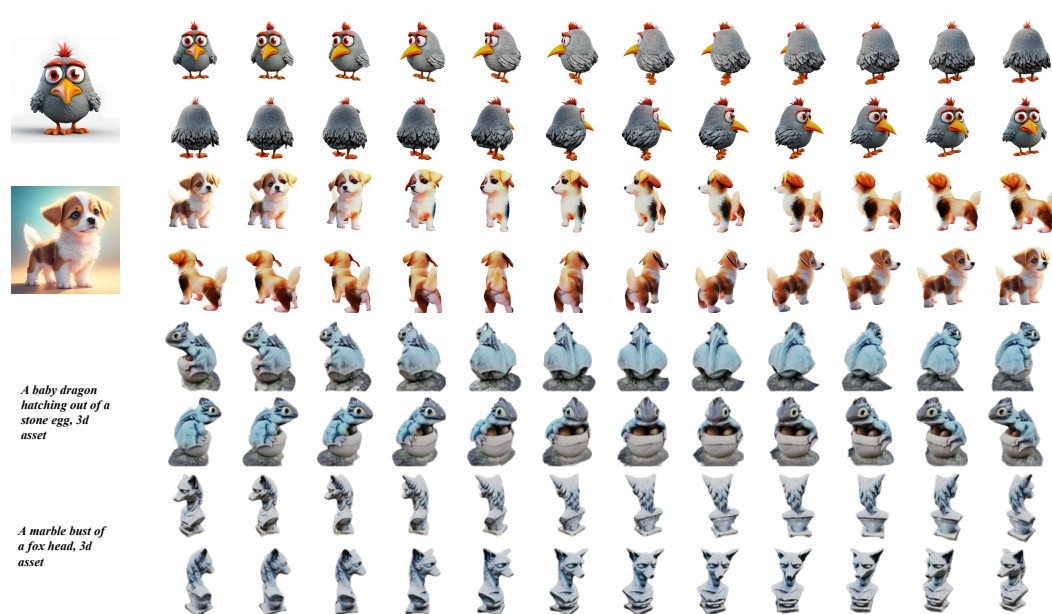

*A baby dragon hatching out of a stone egg, 3d asset*

*A marble bust of a fox head, 3d asset*

Figure 1: Visualization of Image-based and Text-based dense views generation.

et al. (2023). Third, frames in a video are projected from different views of a 3D scene, such that these frames follow an underlying 3D model and present continuous and gradual changes, making it easier to learn cross-frame patterns that enforce multi-view consistency. VideoMV introduces a novel approach to fine-tune a pre-trained video generative model for dense multi-view generation. Only a small high-quality 3D dataset is used. We show how to connect multi-view images of objects with object-centric videos by adding the camera embedding as a residual to the time embedding for each frame.

Unlike previous work that relies only on the multi-view attention module to enhance multi-view consistency, we propose a novel *3D-Aware Denoising Sampling* procedure to further improve multi-view consistency. Specifically, we employ a feed-forward model conditioned on multi-view images generated by VideoMV to explicitly generate 3D models. Subsequently, these generated 3D models are rendered to the corresponding view and replace the original images produced by VideoMV in the denoising loop. Note VideoMV is also different from RenderDiffusion Anciukevičius et al. (2024), viewset diffusion Szymanowicz et al. (2023c), and DMV3D Xu et al. (2023b), which do not use pre-trained 2D diffusion models due to the structure difference. VideoMV put the 3D rectification part in inference stage, and thus can use pretrained 2D and video model prior to enhance the generalizability for unseen text prompts or input images.

Experimental results show that VideoMV outperforms state-of-the-art multi-view synthesis approaches in terms of both efficiency and quality. For example, MVDream Shi et al. (2023b) consumes 2300 GPU hours to train a 4-view generation model. In contrast, VideoMV, which uses weights from a pre-trained video generation model, only requires 4 GPU hours to train a 24-view generation model. On the other hand, VideoMV outperforms MVDream Shi et al. (2023b) in metrics of image quality and multi-view consistency.

In summary, our contributions are as follows:

- We propose VideoMV, which is fine-tuned from off-the-shelf video generative models, for multi-view synthesis. It exhibits strong multi-view consistency behavior.

- We introduce a novel 3D-aware denoising strategy to further improve the multi-view consistency of the generated images.

- Extensive experiments demonstrate that our method outperforms the state-of-the-art approaches in both quantitative and qualitative results.

## 2 RELATED WORKS

**Distillation-based Generation.** Score Distillation Sampling was first proposed by DreamFusion Poole et al. (2022) to generate 3D models by distilling from pre-trained 2D image generative models without using any 3D data. Fantasia3D Chen et al. (2023) further disentangled the optimization into geometry and appearance stages. Magic3D Lin et al. (2023a) uses a coarse-to-fine strategy for high-resolution 3D generation. ProlificDreamer Wang et al. (2023e) proposes variational score distillation (VSD), which models the 3D parameter as a random variable instead of a constant. CSD Kim et al. (2023) considers multiple samples as particles in the update and distills generative priors over a set of images synchronously. NFSD Katzir et al. (2023) proposes an interpretation that can distillate shape under a nominal CFG scale, making the generated data more realistic. SteinDreamer Wang et al. (2023b) reduces the variance in the score distillation process. LucidDreamer Liang et al. (2023) proposes interval score matching to counteract over-smoothing. HiFA Zhu & Zhuang (2023) and DreamTime Huang et al. (2023b) optimize the distillation formulation. RichDreamer Qiu et al. (2023) models the geometry using a multi-view normal-depth diffusion model, which makes the optimization more stable. RealFusion Melas-Kyriazi et al. (2023), Make-it-3D Tang et al. (2023b), HiFi-123 Yu et al. (2023b), and Magic123 Qian et al. (2023) use multi-modal information to improve generation fidelity. DreamGaussian Tang et al. (2023a) and GaussianDreamer Yi et al. (2023) use an efficient Gaussian Splitting representation to accelerate the optimization process. However, distillation-based generation is time-consuming, as it requires tens of thousands of iterations of the 2D generator and can take hours to generate a single asset.

**Feed-forward-based Generation.** Many approaches attempt to use a neural network to directly learn the 3D distribution by fitting 3D data. OccNet Mescheder et al. (2018) encodes shapes in a function space and infers a 3D structure from various inputs. MeshVAE Tan et al. (2017) also learns a reasonable representation in probabilistic latent space for various applications. 3D-GAN Wu et al. (2016) designs a volumetric generative adversarial network for shape generation from latent space. With the development of differentiable rendering, HoloGAN Nguyen-Phuoc et al. (2019) and BlockGAN Nguyen-Phuoc et al. (2020) learn 3D representation from natural images in an unsupervised manner. To maintain multi-view consistency, some prior work Chan et al. (2020; 2021); Deng et al. (2021); Gu et al. (2021); Niemeyer & Geiger (2020); Xu et al. (2021); Zhang et al. (2022) incorporates implicit 3D representations in generative adversarial networks for 3D-aware generation. GET3D Gao et al. (2022), DG3D Zuo et al. (2023), and TextField3D Huang et al. (2023a) leverage DMTet Shen et al. (2021) for accurate textured shape modeling. Assisted by the development of 2D diffusion models Ho et al. (2020); Rombach et al. (2021), 3D diffusion-based approaches Liu et al. (2023d); Kalischek et al. (2022); Zhou et al. (2021); Luo & Hu (2021); Zeng et al. (2022); Chou et al. (2022); Li et al. (2022); Cheng et al. (2022); Zheng et al. (2023); Nam et al. (2022); Muller et al. (2022); Gupta & Gupta (2023); Shue et al. (2022) use variants of diffusion models for generative shape modeling. Point-E Nichol et al. (2022) and Shap-E Jun & Nichol (2023) expand the scope of the training dataset for general object generation. LRM Hong et al. (2023), PF-LRM Wang et al. (2023c), and LGM Tang et al. (2024) choose to use a deterministic approach to reconstruct from a few views. LEAP Jiang et al. (2023) and FORGE Jiang et al. (2022) focus on generating the 3D model using a few images with noisy camera poses or unknown camera poses. While these approaches are many times faster than distillation-based methods, their quality is limited.

**Novel View Synthesis Generation.** Some other work Sajjadi et al. (2022); Wiles et al. (2019); Chan et al. (2023); Gu et al. (2023); Szymanowicz et al. (2023a); Tseng et al. (2023); Yu et al. (2023a); Zhou & Tulsiani (2022); Suhail et al. (2022) combines a novel view generator with a traditional reconstruction process or a fast neural reconstruction network for 3D generation. ViewFormer Kulhánek et al. (2022) uses transformers for novel view synthesis. 3DiM Watson et al. (2022) is the first to use diffusion models for pose-controllable view generation. Zero123 Liu et al. (2023b) adopts a large pre-trained image generator (StableDiffusion Rombach et al. (2021)), which greatly improves generalizability after fine-tuning on Objaverse Deitke et al. (2022). SyncDreamer Liu et al. (2023c) designs a novel depth-wise attention module to generate consistent 16 views with fixed poses. Consistent123 Lin et al. (2023b) combines 2D and 3D diffusion priors for 3D-consistent generation. Zero123++ Shi et al. (2023a) overcomes common issues such as texture degradation and geometric misalignment. Wonder3D Long et al. (2023) introduces a diffusion model between domains. ImageDream Wang & Shi (2023) proposes global control that shapes the overall layout of the object and local control that fine-tunes the details of the image. iNVS Kant et al. (2023) improves the novel view synthesis pipeline through accurate depth warping. MVDream Shi

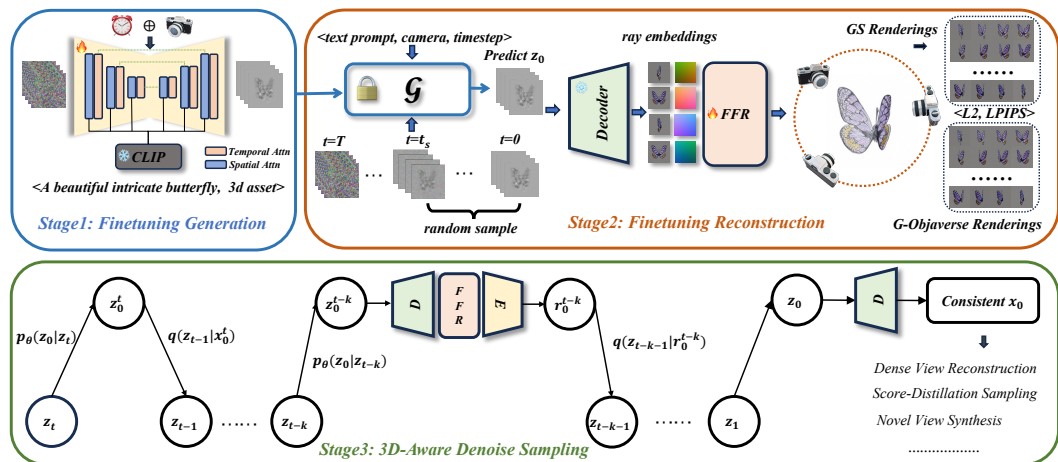

Figure 2: The overall framework. In the first stage, we take a pre-trained video generation model and fine-tune it by incorporating camera poses to generate multi-view images. Then we train a feed-forward reconstruction module(**FFR**) to obtain an explicit global 3D model given noise-corrupted images. Finally, we adopt a 3D-aware denoise sampling strategy that effectively inserts the images rendered from the global 3D model into the denoising loop to further improve consistency.

et al. (2023b) proposes to jointly generate 4 views with dense self-attention on all views. SPAD Kant et al. (2024) further enhances multi-view consistency through proposed epipolar attention.

The concurrent work, IM-3D Melas-Kyriazi et al. (2024) and SVD Blattmann et al. (2023), share a similar idea of generating more consistent multi-view images. The former uses a time-consuming optimization scheme to obtain the 3D model, while the latter adopts the elevation angle instead of the complete camera pose as a condition, posing an obstacle to downstream tasks that require camera pose input. Compared to them, we employ a more efficient feed-forward module to obtain an explicit 3D model from noise-corrupted images. Furthermore, we propose a novel *3D-Aware Denoising Sampling* to further improve consistency.

## 3 METHOD

### 3.1 PROBLEM STATEMENT AND APPROACH OVERVIEW

**Problem Statement.** Given a text or single-image prompt, VideoMV aims to generate consistent multi-view images under user-specified camera poses.

**Approach Overview.** The key idea of VideoMV is to combine a large video generative model for initializing a multi-view generative model and a novel *3D-Aware Denoising Sampling* strategy to further improve multi-view consistency. Figure 2 illustrates the pipeline of VideoMV. In the first stage, we fine-tune a pre-trained video generation model to obtain the multi-view generative model $\mathcal{G}$ (Section 3.2). We focus on how to specify camera poses of multi-view images to connect with object-centric videos. In the second stage, we employ a feed-forward reconstruction module to obtain an explicit global 3D model based on the images generated by $\mathcal{G}$ (Section 3.3). The explicit model uses a variant of the 3D Gaussian splitting (3DGS) representation Kerbl et al. (2023b); Yi et al. (2023); Tang et al. (2024). In the third stage, we introduce a *3D-Aware Denoising Sampling* strategy that effectively inserts the images rendered from the global 3D model into the denoising loop (Section 3.4) to further improve multi-view consistency.

### 3.2 FINE-TUNING GENERATION

The first stage of VideoMV fine-tunes a video generative model for multi-view image generation. This is achieved by generating videos from rendered images of 3D models for fine-tuning. In the following, we first introduce the pre-trained video generative models. We then describe how to generate video data from 3D models for fine-tuning.

**Pre-trained video generative models.** We choose two open-source video generative models, Modelscope-T2V Wang et al. (2023a) and I2VGen-XL Zhang et al. (2023), which are used for

the text-based and single-image-based multi-view generation, respectively. Both belong to the video latent diffusion model (VLDM), which uses a pre-trained encoder and a pre-trained decoder and performs diffusion and denoising in the latent space.

Specifically, consider a video $\boldsymbol{x} \in \mathbb{R}^{F \times H \times W \times 3}$ where $F$ is the number of frames. They use a pre-trained encoder $\mathcal{E}$ of VQGAN Esser et al. (2021) to compress it into a low-dimensional latent feature $\boldsymbol{z} = \mathcal{E}(\boldsymbol{x})$, where $\boldsymbol{z} \in \mathbb{R}^{F \times h \times w \times c}$. In the training stage, the diffusion process samples a time step $t$ and converts $\boldsymbol{z_0}$ to $\boldsymbol{z_t}$ by injecting Gaussian noise $\epsilon$. Then a denoising network $\epsilon_\theta$ predicts the added noise $\epsilon_\theta(\boldsymbol{z_t}, y, t)$. The corresponding optimized objective can be simplified as follows:

$$\mathcal{L}_{\text{VLDM}} = \mathbb{E}_{\boldsymbol{z_t}, y, \epsilon \in \mathcal{N}(0,1), t} \|\epsilon - \epsilon_\theta(\boldsymbol{z_t}, y, t)\|_2^2, \tag{1}$$

where $y$ denotes the conditional text or image. In the denoising sampling loop, given an initial Gaussian noise, the denoising network predicts the added noise $\epsilon_\theta(\boldsymbol{z_t}, y, t)$ for each step, ultimately obtaining a latent code $\boldsymbol{z_0}$, which is fed into the decoder of VQGAN Esser et al. (2021) to recover a high-fidelity video.

**Video data generation for fine-tuning.** We utilize the 3D G-Objaverse data-set Qiu et al. (2023) to generate video data, denoted as $\boldsymbol{x}$, to fine-tune the video generation model. A key challenge is to generate data that is suitable for downstream tasks of multi-view image generation but does not present a large domain gap to the pre-trained video generation model. To this end, we generate a video of rendered images by rotating the camera around each 3D object in the G-Objaverse dataset Qiu et al. (2023). In our experiment, we select 24 views for each object with a fixed elevation angle (randomly selected from 5 to 30 degrees) and azimuth angles uniformly distributed between 0 and 360 degrees.

Note that VLDM uses efficient temporal convolution and attention, which operate at the same positions between frames. This is very different from the dense attention mechanism used in MV-Dream Shi et al. (2023b), which operates at all positions between frames, making memory explosion for dense views generation. To utilize VLDM for fine-tuning, dense views work much better than sparse views. On the other hand, dense views offer more flexibility for downstream tasks.

VideoMV also uses camera poses as an additional control to generate images of different viewpoints, which support arbitrary novel view synthesis. Inspired by previous work Shi et al. (2023b); Liu et al. (2023c); Long et al. (2023), we use a two-layer multi-layer perception (MLP) to extract a camera embedding, which is combined with the time embedding. In other words, the noise predicted by the denoising network changes to $\epsilon_\theta(\boldsymbol{z_t}, y, c, t)$, where $c$ denotes the camera poses. Furthermore, to maintain the generalizability of our model, we integrate additional 2D image data from LAION 2B Schuhmann et al. (2022). These images are treated as videos with the number of views set to 1. After fine-tuning, we obtain a diffusion model, which outputs multiview images conditioned text or a single image.

### 3.3 FEED-FORWARD RECONSTRUCTION

The second stage of VideoMV learns a neural network that reconstructs a 3D model from images generated by the model $\mathcal{G}$ trained in the first stage. In the last stage of VideoMV, we will use rendered images of this 3D model to guide the denoising step in $\mathcal{G}$ to achieve improved multi-view consistency.

We employ 3D Gaussians Kerbl et al. (2023b) as the representation of the 3D model, which has a fast rendering pipeline for image generation. Instead of using the optimization scheme that gets 3D Gaussians parameters via fitting rendering images to input images (which is time-consuming), we employ a feed-forward manner to directly regress the attributes and number of 3D Gaussians. In the following, we first review the 3D Gaussian Splatting Kerbl et al. (2023b) representation. We then present the reconstruction network.

**3D Gaussians.** The 3D Gaussian representation uses a set of 3D Gaussians to represent the underlying scene. Each Gaussian is parameterized by a center $\mathbf{p} \in \mathbb{R}^3$, a scaling factor $\mathbf{s} \in \mathbb{R}^3$, a rotation quaternion $\mathbf{q} \in \mathbb{R}^4$, an opacity value $\alpha \in \mathbb{R}$, and a color feature $\mathbf{c} \in \mathbb{R}^C$. To render the image, 3DGS projects the 3D Gaussians onto the camera imaging plane as 2D Gaussians and performs alpha compositing on each pixel in front-to-back depth order.

**Reconstruction network.** Inspired by splatter image Szymanowicz et al. (2023b) and LGM Tang et al. (2024), we first designed a reconstruction network that learns to convert noise-corrupted multi-view latent features in the denoising procedure of $\mathcal{G}$ into Gaussian correlation feature maps, whose channel values represent the parameters of the Gaussian and whose number of pixels is equal to the 3D Gaussian number. However, we find this module difficult to learn, causing the rendered images to become blurred. One explanation is that the latent space is highly compressed, and it is difficult to learn patterns between this latent space and the underlying 3D Gaussian model. To address this issue, we adopt the decoder of VQGAN Esser et al. (2021) to decode the noise latent features into images and use these images as input for this module. For reconstruction, we employ LGM Tang et al. (2024) and its powerful pre-trained weights for fast training convergence. Furthermore, following LGM Tang et al. (2024) and DMV3D Xu et al. (2023a), we use Plücker ray embeddings to densely encode the camera pose, and the RGB values and ray embeddings are concatenated together as input to this reconstruction module.

The task of this network is to recover global 3D even if the input multi-view images are noise-corrupted or inconsistent. Unlike LGM Tang et al. (2024), which uses data augmentation strategies to simulate inconsistent artifacts of input multi-view images, we directly use the output of our multi-view generative model $\mathcal{G}$ to train the reconstruction model. In this way, we do not encounter domain gaps between the training and inference stages. Specifically, we train this network using the noise-corrupted images obtained by only a single denoising step of $\mathcal{G}$. The original output of $\mathcal{G}$ is the predicted noise according to the input time step $t \in [0, 1000]$, and we convert it to noise-corrupted multi-view images as training data. The details of conversion will be introduced in the next Section 3.4. In the larger timestep, the converted multi-view images are similar to Gaussian noise, which is not suitable as training data for the reconstruction network. Therefore, we select time steps in the range of $[0, t_s]$ (we set $t_s = 700$) to train our module.

## 3.4 3D-Aware Denoising Sampling

As shown in Figure 2, we adopt a *3D-Aware Denoising Sampling* strategy that involves the rendered images produced by our reconstruction module in a denoising loop to further improve the multi-view consistency of the resulting images. We use the DDIM Song et al. (2020) scheduler with 50 denoised steps for fast sampling. The sampling step from $z_t$ to $z_{t-1}$ of DDIM Song et al. (2020) can be formulated as follows:

$$z_{t-1} = \sqrt{\alpha_{t-1}} \underbrace{\left( \frac{z_t - \sqrt{1-\alpha_t}\epsilon_\theta^{(t)}(z_t)}{\sqrt{\alpha_t}} \right)}_{\text{`` predicted } z_0\text{''}} + \underbrace{\sqrt{1-\alpha_{t-1}-\sigma_t^2} \cdot \epsilon_\theta^{(t)}(z_t)}_{\text{``direction pointing to } z_t\text{''}} + \underbrace{\sigma_t\epsilon_t}_{\text{random noise}} , \quad (2)$$

where $\alpha_t$ and $\sigma_t$ are constants, $\epsilon_t$ is the standard Gaussian noise independent of $z_t$, and we use $\epsilon_\theta^{(t)}$ rather than $\epsilon_\theta(z_t, y, c, t)$ to denote the predicted noise for simplicity. Note that during the training of the reconstruction network, we convert the predicted noise to "predicted $z_0$" and decode it to $x_0$ as the input of the training data.

Table 1: Quantitative comparison of text-based multi-view generation: Our proposal achieves consistently better performance in both dense views (f=1) and sparse views (f=6) settings.

| Method | PSNR↑ | SSIM↑ | LPIPS↓ | ClipS↑ | RMSE(f=1)↓ | RMSE(f=6)↓ | Points↑ |
|---|---|---|---|---|---|---|---|
| MVDream | 20.50 | 0.6708 | 0.4156 | 35.33 | 0.0637 | 0.0969 | 133 |
| VideoMV | **23.32** | **0.7638** | **0.3682** | **35.45** | **0. 0536** | **0.0948** | 1650 |

Table 2: Quantitative comparison of image-based multi-view generation.

| Method | PSNR↑ | | SSIM↑ | | LPIPS↓ | | RMSE↓ | | CD ↓ | IOU↑ |
|---|---|---|---|---|---|---|---|---|---|---|
| Zero123 | 15.36 | | 0.773 | | 0.1689 | | 0.1404 | | 0.0373 | 0.4521 |
| Zero123-XL | 15.82 | | 0.778 | | 0.1622 | | 0.1417 | | 0.0354 | 0.4846 |
| SyncDreamer | 16.88 | | 0.790 | | 0.1589 | | 0.1368 | | 0.0278 | 0.5156 |
| VideoMV | **18.24** | | **0.809** | | **0.1433** | | **0.1278** | | **0.0257** | **0.5228** |
| Views | 4 | 24 | 4 | 24 | 4 | 24 | 4 | 24 | | |
| ImageDream | 11.84 | 11.41 | 0.7256 | 0.7210 | 0.3239 | 0.3367 | **0.1037** | **0.0670** | 0.0519 | 0.3974 |
| VideoMV | **20.02** | **17.09** | **0.8200** | **0.7978** | **0.1382** | **0.1532** | 0.1490 | 0.0759 | **0.0257** | **0.5228** |

In the denoising sampling loop, we employ the more consistent "reconstructed $z_0$" to participate in the loop, where the "reconstructed $z_0$" is rendered by our reconstruction module by passing "predicted $z_0$". However, this process involves decoding $z_0$ to $x_0$ and encoding $x_0$ to $z_0$, which may

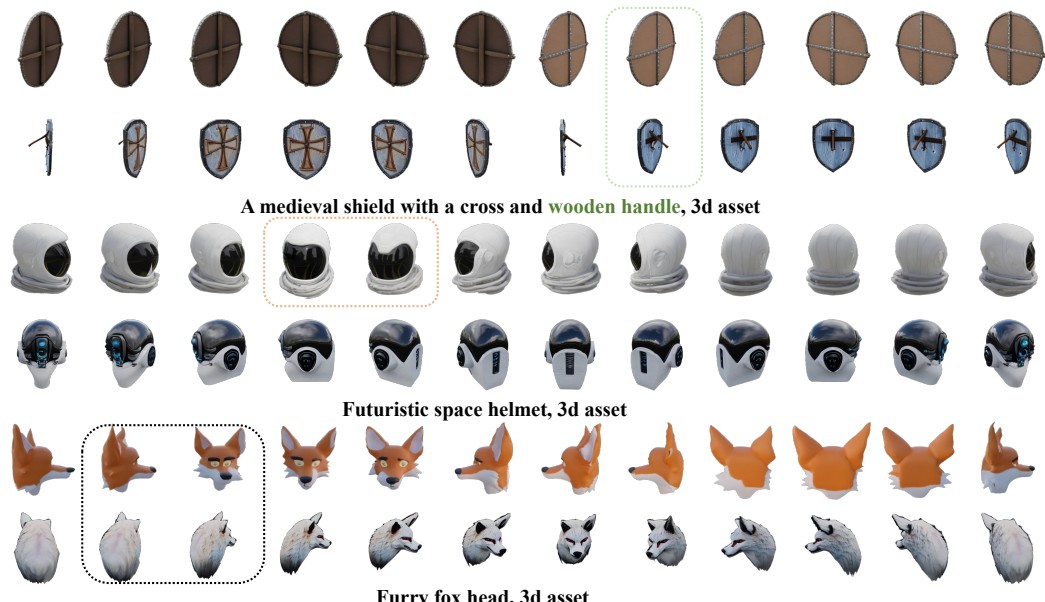

Figure 3: Qualitative comparison of MVDream Shi et al. (2023b) (**Up**) and VideoMV (**Down**) VideoMV can generate high-fidelity multi-view images which align to the text description with accurate camera control and consistent content. However, MVDream easily suffered from **inaccurate pose control** and content drifting.

encounter efficiency problems. To address this issue, we use a simple strategy of using "reconstructed $z_0$" every $k$ timestep (we set $k = 10$). We also skip it in the early denoising step. This is also reasonable since the predicted images are noising in the early steps, and thus there is no need to reconstruct.

In addition to generating multi-view images after the denoising loop, we also obtain a global 3D model represented by 3D Gaussians. We can convert the 3D Gaussians into a polygonal mesh, i.e., by training an efficient NeRF Mildenhall et al. (2020); Wang et al. (2021; 2023d) from rendered images of 3D Gaussians and extracting a mesh from the density field of the resulting NeRF.

## 4 EXPERIMENTS

We perform experimental evaluation on two tasks, i.e., text-based multi-view generation and image-based multi-view generation. For text-based multi-view generation, we adopt MVDream Shi et al. (2023b) as the baseline approach, and report metrics including PSNR, SSIM Wang et al. (2004), LPIPS Zhang et al. (2018), and flow-warping RMSE. For image-based multi-view generation, we adopt Zero123 Liu et al. (2023b), Zero123-XL Deitke et al. (2024); Liu et al. (2023b), and SyncDreamer Liu et al. (2023c) as baseline approaches, and report metrics that include PSNR, SSIM Wang et al. (2004), and LPIPS Zhang et al. (2018). Note that in text-based multi-view generation, we evaluate by NeRF-based novel view synthesis since no ground truth is provided.

### 4.1 TEXT-BASED MULTI-VIEW GENERATION

We use 100 single-object prompts from T3Bench He et al. (2023) for quantitative evaluation. For MVDream Shi et al. (2023b), we feed circular camera poses into it and generate 24 views simultaneously. MVDream Shi et al. (2023b) was trained on 32 uniformly distributed azimuth angles, and the objects were rendered twice with different random settings. Therefore, MVDream is able to generate more views interpolately given the camera poses. VideoMV was trained at 24 uniformly distributed azimuth angles, and the objects were rendered only once with random elevations. (G-Objaverse Qiu et al. (2023)). After we generate 24 views by a specific text prompt, we use 12 views for a neural field reconstruction(instant-ngp) and report the novel view synthesis metrics (PSNR, SSIM Wang et al. (2004), and LPIPS Zhang et al. (2018)) on the remaining 12 views to evaluate the multi-view consistency. We also calculated the average Clip-Score between the text prompt and generated 24 views to assess the text-to-image alignment. Another metric is flow-warping RMSE Liu

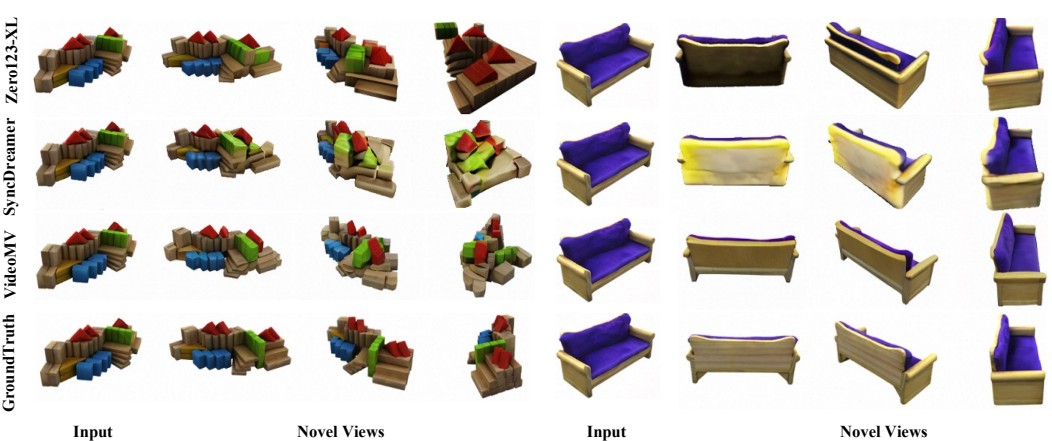

Figure 4: Image-based multi-view generation on GSO Downs et al. (2022) test dataset(**First column as the input view**).

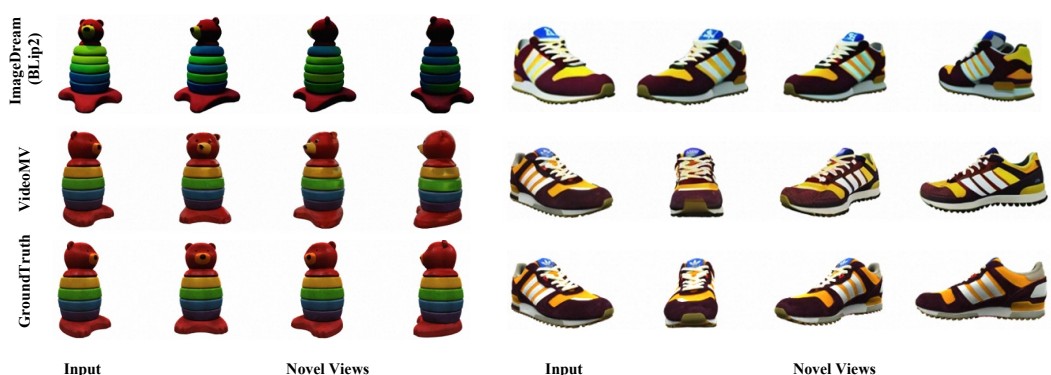

Figure 5: Comparison with ImageDream on GSO Downs et al. (2022) test dataset(**First column of GroundTruth as the input view**).

et al. (2023a), which is widely adopted in 3D and video editing to evaluate semantic consistency between short-ranged or long-ranged frames. We use RAFT Teed & Deng (2020) for optical flow estimation and softmax-splatting for warping between consecutive frames. We report Flow-Warping RMSE on two settings: one with an interval of every 1 frame and the other with an interval of every 6 frames. Note an interval of 6 frames aligns perfectly with MVDream since it is trained to produce 4 orthogonal views. We do not use 4 views for novel view synthesis evaluation since 4 views are two sparse for a reconstruction pipeline. To avoid the ambiguity that we use output of NeRF as a pseudo ground truth, we also report the SfM Schönberger & Frahm (2016) result of all 24 views.

As depicted in Tab. 1, VideoMV significantly outperforms MVDream in 3D consistency-related metrics (PSNR, SSIM Wang et al. (2004), LPIPS Zhang et al. (2018)) and flow-warping RMSE using an interval of every 1 frame. VideoMV achieves a similar Clip-Score although trained with less data and a slightly better flow-warping RMSE using an interval of every 6 frames, demonstrating the effectiveness of *3D-Aware Denoising Sampling* guided by an underlying 3D model. The reconstructed point number again makes up for the potential insufficient views for NeRF reconstruction and futher verify the effectiveness of our proposal.

Due to space constraints, we visualize some typical results with only 12 views in Fig. 3 for qualitative comparison with MVDream Shi et al. (2023b). We refer the readers to the supp. material for a visualization with all 24 views. Although trained with 4 views with random angles simultaneously, MVDream Shi et al. (2023b) still suffered from content drifting and inaccurate pose control. In contrast, VideoMV can provide precise camera control without content drifting over dense views. VideoMV can provide consistent and fine-grained dense-view prior for downstream tasks like dense view reconstruction and distillation-based 3D generation.

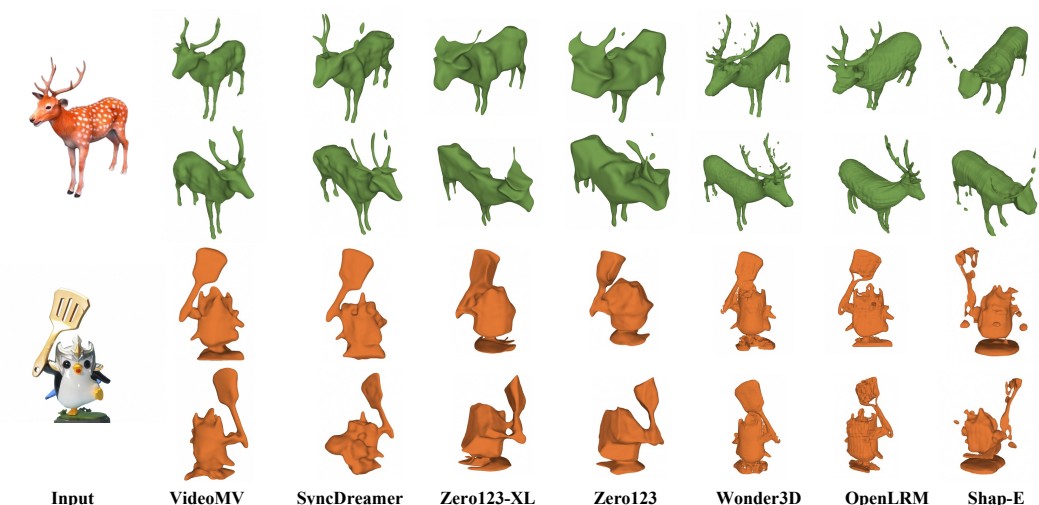

| Input | VideoMV | SyncDreamer | Zero123-XL | Zero123 | Wonder3D | OpenLRM | Shap-E |

Figure 6: Image-based 3D generation results from NVS-based methods and inference-based methods.

## 4.2 IMAGE-BASED MULTI-VIEW GENERATION

VideoMV can be reformulated to image-based multi-view generation. In our experiments, we fine-tune VideoMV from I2VGen-XL Zhang et al. (2023), which is an open-source image-based video generation method and shares the same architecture as modelscopeT2V Wang et al. (2023a). Since I2VGen-XL Zhang et al. (2023) accepts both an input image and a text prompt, we set the text prompt to an empty string in the fine-tuning stage. We similarly train a feed-forward reconstruction module and apply consistent sampling in VideoMV. Evaluation is carried out on 50 objects from the GSO dataset (Google Scanned Objects) dataset Downs et al. (2022), including the 30 objects from SyncDreamer Liu et al. (2023c). Since SyncDreamer Liu et al. (2023c) is trained to generate fixed 16 views with an elevation of 30 degrees and the azimuth spans evenly in [0, 360] degrees, we only compute metrics on the $[0, 3, 6, 9, 12, 15, 18, 21]^{th}$ frames that correspond to the $[0, 2, 4, 6, 8, 10, 12, 14]^{th}$ frames of SyncDreamer Liu et al. (2023c). For Zero123 Liu et al. (2023b) and Zero123-XL Deitke et al. (2024), we report metrics on the generated frames with azimuths of $[0, 45, 90, 135, 180, 225, 270, 315]$ degrees. We also compare our method with ImageDream Wang & Shi (2023), which is an image-prompt-based multi-view generation method. We use BLIP2 Li et al. (2023) to caption the input image and evaluate under settings of 4 views and 24 views, respectively. Note that we always evaluate under the elevation settings of our baselines for fairness, which means that we use an input image of $elevation = 5$ for ImageDream Wang & Shi (2023) and an input image of $elevation = 30$ for SyncDreamer Liu et al. (2023c).

We first visualize some image-based multi-view generation results among our testing GSO dataset Downs et al. (2022) in Fig. 4. Zero123 and Zero123-XL Liu et al. (2023b) suffer content drift since no global 3D information is utilized. SyncDreamer Liu et al. (2023c) generates geometry-consistent multi-view images with coarse colors due to the discrete depth-wise attention applied to the low-resolution latent space. VideoMV generates more consistent results with precise colors since it adopts a global 3D representation in the full-resolution image space and utilizes the strong multi-view prior from large video generative models. The numerical results in Tab. 2 also consistently align with the visualization results. We find that ImageDream obtains significantly lower PSNR, SSIM, and LPIPS, but achieves better flow-warping RMSE under different settings of views. To clarify this, we also visualize the novel views generated by ImageDream in Fig. 5. As depicted, ImageDream generates novel views based on the input image and text prompt, which produces prompt-aligned multi-view images but does not consistently follow the pixel-level constraint of the input image. Moreover, it also suffers from inaccurate pose control and content drifting problems since it is based on MVDream Shi et al. (2023b). It achieves better flow-warping RMSE in the 24 views setting, since it sometimes produces consecutive images with the same pose(see the samples of ImageDream in Fig. 5). Despite these shortcomings, ImageDream maintains better semantic consistency under the 4 views setting, which makes it more suitable for distillation sampling than VideoMV.

Inspired by prior work, we present Volume IOU and Chamfer Distance metrics on the GSO dataset using the off-the-shelf MVS method, such as NeuS Wang et al. (2021). As depicted in Tab. 2, VideoMV outperforms state-of-the-art methods in terms of Chamfer Distance and Volume IOU metrics, indicating that leveraging increased consistency in multi-view images for reconstruction can result in improved accuracy in 3D geometry.

## 4.3 ABLATION STUDY

Video models are easy to get object-centric prior due to its training dataset containing abundant and diverse videos. We use multi-view images to distillate this prior knowledge and ensure the multi-view consistency by inherent spatial-temporal attention module and our proposed novel 3D-Aware sampling, which makes VideoMV different from previous methods based on image models. Related metrics depicted in Tab. 3 also shows the performance drop if we **zero out the temporal attention layer**(prior from videos) when we load the pre-trained weight of video models. **'base'** denotes that we do not apply 3D-aware denoise sampling in the inference stage.

Table 3: Quantitative results of various ablation settings.

|  | PSNR | SSIM | LPIPS | ClipS |
|---|---|---|---|---|
| VideoMV(**base**, **zeroing out**) | 20.09 | 0.6593 | 0.4228 | 33.77 |
| VideoMV(**base**) | 22.92 | 0.7551 | 0.4107 | 35.47 |
| VideoMV | **23.32** | **0.7638** | **0.3682** | **35.45** |

A silver helmet, 3d asset

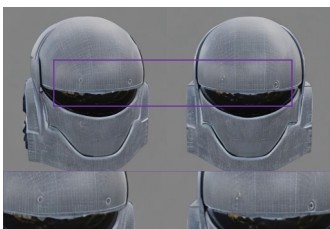
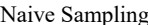
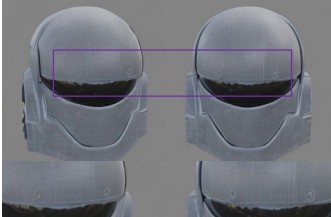

Naive Sampling          3D-Aware Denoise Sampling

Figure 7: Ablation for *3D-Aware Denoising Sampling*.

Observing the changes in images before and after applying *3D-Aware Denoising Sampling* reveal a clear increase of consistency in the human vision system. As shown in Fig. 7, starting from same initial noise, *3D-Aware Denoising Sampling* significantly improves the consistency of novel views over baseline (Naive Sampling). This shows the effectiveness of our proposed *3D-Aware Denoising Sampling* strategy.

## 5 CONCLUSIONS

In this paper, we present a consistent dense multi-view generation method that can generate 24 views at various elevation angles. By fine-tuning large video generative models for several GPU hours, our proposal can effectively produce dense and consistent multi-view images from an input image or a text prompt. Future directions may focus on developing a robust neural reconstruction pipeline based on the provided consistent dense views. Moreover, we have shown that there are rich opportunities in connecting videos and multi-view based 3D vision tasks. We hope our findings in turning a video generative model into a consistent multi-view image generator can also inspire other 3D generation and video-related tasks.

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
