# OpenReview forum: "VideoMV: Consistent Multi-View Generation Based on Large Video Generative Model"
_ICLR.cc/2025/Conference — ICLR 2025 Conference Withdrawn Submission_

### Official Review · Reviewer_mNus · 2024-10-28

**Soundness:** 2
**Presentation:** 2
**Contribution:** 2
**Rating:** 3
**Confidence:** 4

**Summary:**

This paper investigates 3D content creation by generating multi-view images. Specifically, it generates multi-view images based on text or image prompts and uses the generated images to reconstruct 3D representation. As claimed in the paper, with pre-trained video generation models, the proposed method achieves multi-view image generation with fast convergence speed.

**Strengths:**

1. This paper propose a method for both image-based and text-based 3D generation.

2. The proposed framework achieves good performance.

3. New metrics are introduced to evaluating 3D consistency.

**Weaknesses:**

1. **The motivation is unclear.** First, as claimed in the introduction section, one of the motivations for this paper is the scarcity of large-scale 3D data. However, despite lagging behind the 2D data, the amount of 3D data is still quite large, with 10M objects in Objaverse-XL. In my view, the lack of large-scale 3D data cannot be a clear motivation for this paper.
Second, increasing the convergence speed and improving the quality can be a reasonable motivation for using pre-trained video diffusion models. However, despite viewing it as the contribution, this paper does not provide the details regarding the training data and training cost.

2. **Fine-tuning a video generation model for 3D generation is not novel.** Leveraging the implicit knowledge of video generation models to help 3D generation has been investigated in recent works (e.g., SV3D and IM-3D) and achieves significant performance. In Sec. 2, this paper introduces IM-3D and SVD (not sure if it is a typo, but I feel it should be SV3D) but does not provide a clear comparison with them. This paper claims that IM-3D requires a time-consuming optimization scheme to obtain a 3D model. However, if exporting mesh is required, this paper also needs per-sample optimization. In addition, this paper considers the elevation angle condition of SVD is limited. However, despite using both elevation and azimuth conditions, this paper evenly split the 360 degree into 24 frames. To this end, I doubt whether there is difference or not when taking azimuth into conditions.
Therefore, I feel the novelty of this paper is limited and it also does not comprehensively discuss related works.


3. **More explanation about reconstruction network.** Reconstruction network is another main contribution of this paper, which is a feed-forward network predicting gaussians from the intermediate images in the denoising process. However, the predicted gaussians are not used as the final 3D representation but as an intermediate feature. I hope the authors could explain more about the motivation of this. In addition, the reconstruction network is used in the 3D-aware denoising sampling to replace the denoising images with the rendered images. I am curious if this replacement is still under the constraints of DDIM. It is better to theoretically prove it.


4. **The evaluation metrics cannot fully represent the performance of 3D generation and the comparison is unfair.** As far as I know, this paper introduces a new set of metrics for evaluation. It uses half of its generated views to train a NeRF model and calculate the PSNR, SSIM and LPIPS between the other generated views and rendered views. These metrics are interesting and can represent the 3D consistent among the generated views. However, the authors evaluate the MVDream by generating 24 frames instead of 32 frames the model was trained with, which might impair the performance of MVDream. In addition, for fair comparison, the authors should provide the evaluation results on the public benchmarks (T3Bench for text-to-3D and ImageDream Benchmark for image-to-3D).

**Questions:**

My concerns are mainly about the motivation of the method, novelty, more explanation and the evaluation. Please refer to weaknesses.

---

### Official Review · Reviewer_FUfB · 2024-10-30

**Soundness:** 2
**Presentation:** 2
**Contribution:** 2
**Rating:** 5
**Confidence:** 5

**Summary:**

This paper proposes a framework for the multi-view generation task utilizing video generative models. The introduced 3D-aware denoising strategy aims to enhance multi-view consistency in object reconstruction. Experiments have been conducted to demonstrate the method’s effectiveness and superiority over existing approaches.

**Strengths:**

1. The paper shows its effectiveness over SOTA methods in both qualitative and quantitative ways.
2. The paper can generate not only multi-view images but also the object's 3D model.

**Weaknesses:**

1. The paper lacks novelty.
2. The presentation and organization of the paper should be refined.
3. The paper lacks necessary experiments on evaluation and ablation.

**Questions:**

1. The current novelty of this paper is limited. Although the proposed 3D-aware denoising strategy is intended as the primary innovative contribution, the guided denoising approach, which substitutes the denoised result with the output from the GS, appears overly simplistic. This approach currently lacks the strategic depth necessary to be considered truly novel. Moreover, as indicated in Table 3, the 3D-aware denoising strategy only improves the model by 0.4 dB in PSNR and 0.009 in SSIM. These incremental gains appear minimal, which raises questions about the overall effectiveness and necessity of this approach. It may be beneficial to refine the 3D-aware denoising method for more substantial improvements.

2. The authors should consider conducting more ablation studies to substantiate the effectiveness of their approach. For instance, to validate the impact of the FFR in stage 2, a comparative experiment could be performed where a 3D Gaussian is directly optimized based on images generated by the fine-tuned generative model. The 3D-aware denoising strategy in the inference phase can also be included.

3. The paper lacks reported metrics for evaluating pose accuracy, which are crucial for validating the claims made. The authors mention that "MVDream easily suffers from inaccurate pose control and content drifting," yet there is no quantitative evidence to support this observation. Providing qualitative metrics, such as pose error using a tool like Evo, would strengthen the paper by offering concrete measures of pose accuracy and helping to substantiate the noted limitations in pose control.

4. The organization could benefit from refinement. For instance, Tables 1 and 2 appear on page 6, yet they seem more appropriate within the experimental section. Additionally, the captions for these tables are too brief, making it difficult to interpret the results fully, especially in the second part of Table 2, where the comparison between ImageDream and VideoVM lacks clarity. While the authors provide supplementary results, the in-paper comparison appears limited, and expanding on these comparisons in the main text would help strengthen the reader's understanding of the findings. For example, as shown in Fig. 7, the visual differences are minimal, making it difficult to observe any significant improvement from the proposed method.

---

### Official Review · Reviewer_NxAT · 2024-11-01

**Soundness:** 2
**Presentation:** 2
**Contribution:** 2
**Rating:** 5
**Confidence:** 4

**Summary:**

In this paper the authors propose a model named VideoMV for multi-view generation from single or sparse views input. And further design modules to use generated views for 3D reconstruction.

The VideoMV model is consist of three parts: first part is a fine-tuned video diffusion model that outputs 24 views given text or image conditions. The second part is a feed-forward reconstruction model that fine-tuned together with the video diffusion model, and directly output 3D gaussian parameters for the object content. Last part is a 3D-aware denoise sampling stage that refine the reconstructed gaussian representation.

Overall experiments show satisfactory results and evaluation metrics comparing to previous works.

**Strengths:**

1. The overall results is better than previous methods. And extensive comparison on various benchmarks show the generalization ability of the proposed VideoMV model.

2. The supplementary provides a detailed introduction of the model architecture and implementation details, should be useful for further and make the overall pipeline more straightforward to understand.

3. As authors mentioned, the proposed VideoMV, as inheriting lots of strong implicit prior from previous model, only requires a short period of time on the fine-tuning to achieve better generation quality.

**Weaknesses:**

1. The overall presentation of the work seems to be very confusing. According to the title the VideoMV should be a video generation based model for multiple view synthesis, but then it seems the large portion of the paper is introducing the feed-forward reconstruction and 3D-aware denoising sampling stage. It's fine to have different parts of the method but maybe changing the title accordingly.

2. I think the usage of both feed-forward reconstruction and 3D-aware denoising sampling stage is not well motivated. Also the presentation in the paper on introducing these modules needs to be further polished. Currently there exists many confusing terms such at "Gaussian correlation feature maps" that only appear once without enough explanation.

3. The paper seems to be written in a rush. All references are not highlighted and don't support jumping to sources.

**Questions:**

1. As stated in the weakness part, not sure if the authors are trying to frame all three parts into a video generation framework to match the paper title, if so, the overall writing needs to be largely improved.

2. In the third stage, 3D-aware denoising sampling, what is the diffusion model it's using to get the gradient supervision? Not clear if it's the fine-tuned video generation model.

3. If it's the video generation model itself, what's the main motivation and advantage of this step? Is it just a few steps refinement with a consistent 3D representation comparing to previous two steps? So the consistency issue should be further improved?

---

### Official Review · Reviewer_BPpx · 2024-11-03

**Soundness:** 2
**Presentation:** 3
**Contribution:** 2
**Rating:** 5
**Confidence:** 5

**Summary:**

This paper introduces VideoMV for multi-view 3D content generation. The key insight is to fine-tune existing video generation models for multi-view generation, leveraging the inherent frame consistency in video models. Specifically, VideoMV consists of three stages: 1) fine-tuning a video generation model for multi-view generation; 2) training a feed-forward reconstruction module for explicit 3D modeling; 3) proposing a 3D-aware denoising sampling strategy to enhance multi-view consistency. Experimental results show that VideoMV outperforms the selected existing methods in both generation quality and training efficiency.

**Strengths:**

1. The paper is well-written and structured, with clear presentation of the methodology and results.
2. The overall pipeline is comprehensive and elegant, integrating both multi-view image generation and 3D Gaussian reconstruction in a unified framework.
3. The proposed 3D-aware denoising sampling strategy is well-motivated and effective in improving multi-view consistency.

**Weaknesses:**

1. The comparison with baseline methods is incomplete. Specifically, there are several existing works that also fine-tune video diffusion models for multi-view generation, such as SV3D [1] and VFusion3D [2]. The lack of those baselines make me not convinced about the superiority of the proposed methods in the experiment part.

2. Once again, if we take paper like SV3D into consideration, the author need to prove their proposed 3D-aware denoising sampling strategy --- which I believe is the most novel part in this paper is somehow compariable with the fine-tuning stragtegy in SV3D.

3. Given that the reconstruction network is fine-tuned from LGM, a critical baseline comparison with MVDream+LGM is notably absent in Figure 6.

4. The paper's emphasis appears misaligned with its claimed contributions. If the main novelty lies in the fine-tuned video diffusion model rather than the adapted reconstruction network, the experimental results should focus more on comparing multi-view generation capabilities instead of reconstruction results

5. It is also unfair to compare with single-view reconstruction methods like Open-LRM



[1] Sv3d: Novel multi-view synthesis and 3d generation from a single image using latent video diffusion
[2] Vfusion3d: Learning scalable 3d generative models from video diffusion models

**Questions:**

Why reconstruction metrics such as PSNR, Lpips can be considered as 3D consistency metrics as stated in the manuscripts?

---

### Official Review · Reviewer_c61N · 2024-11-04

**Soundness:** 2
**Presentation:** 1
**Contribution:** 2
**Rating:** 3
**Confidence:** 3

**Summary:**

The paper proposes a multi-stage pipeline for finetuning a video generation model for multivew generation. In the first stage, the video model is trained on posed object-centric videos. In the second stage, a feedforward 3D reconstruction model is trained based on the rendered pseudo-GT multiview examples sampled from the first stage model. In the final stage, the feedforward reconstruction is used as initialization for a 3D-aware denoising of the final multiview images, which improves their 3D consistency.

**Strengths:**

The qualitative and quantitative results are visually impressive on the given view synthesis benchmarks.

The fact that outputs of a fine tuned multiview diffusion model are consistent enough to be used as pseudo-GT for a feedforward 3D model similar to Splatter Image is notable and could be valuable especially if validated more experimentally and qualitatively.

**Weaknesses:**

The technical novelty is limited. The basic idea is to fine-tune a VDM with pose-conditioning on Objaverse, then train a Splatter Image like model, and finally apply a 3D-aware resampling-like procedure to generate consistent views similar to ReconFusion (which should probably be cited).

The paper claims in various places that the second stage can be used to perform feedforward reconstruction of Gaussian Splats. However, given that this would likely be based on camera poses from a camera conditioned video model, the alignment would be relatively imprecise resulting in blurry reconstructions. Indeed, I cannot find any examples of the reconstructed splats from Stage 2 in the main paper or supp, though some distillation results are presented, but this is not the feedforward model as claimed. It would be good to either separately experimentally validate and qualitatively show the results of the Gaussian Splatting model in the second stage, or tone down the claims of feedforward 3D reconstruction leaving out the third stage, which undoubtedly refines many details.

Frequent formatting/writing issues such as:
incorrect use of \citet, \citep
“3D gaussian splitting” -> splatting
Stylization of methods and paper titles: “splatter image” should be “Splatter Image”, etc.

**Questions:**

Can you please show results from Stage 2 alone (reconstructed gaussian splats from the feedforward model) and possibly also quantitatively analyze them? This will help to contextualize the claims in the paper about feedforward 3D reconstruction, and performance/runtime.

---

### Note · Authors · 2024-11-12

I have read and agree with the venue's withdrawal policy on behalf of myself and my co-authors.